# Effect of *Aloe vera* Gel as a Natural Antioxidant on the Quality of Cold-Stored Sea Bass (*Dicentrarchus labrax*)

**DOI:** 10.3390/foods14071185

**Published:** 2025-03-28

**Authors:** Fachruqi Waris, Mutlu Pilavtepe-Celik

**Affiliations:** 1Program of Fisheries, Graduate School of Natural and Applied Sciences, Kocaeli University, Kocaeli 41001, Türkiye; fachruqiukhy@gmail.com; 2Department of Food Processing, Izmit Vocational School, Kocaeli University, Kocaeli 41285, Türkiye; 3Department of Chemical Engineering, Faculty of Engineering, Gebze Technical University, Kocaeli 41400, Türkiye

**Keywords:** *Aloe vera*, natural antioxidant, sea bass, lipid oxidation, cold storage, shelf life

## Abstract

Growing health and environmental concerns have intensified the search for natural antioxidants to replace synthetic alternatives to prevent spoilage of seafood. Long-term intake of synthetic antioxidants has been linked to liver toxicity, reproductive issues, and cancer risks in animals. This study therefore evaluates the efficacy of *Aloe vera* gel (AVG) as a natural antioxidant in preserving the quality of sea bass (*Dicentrarchus labrax*, Linnaeus 1758) slices during cold storage at 4 ± 1 °C for 13 days. Sea bass slices were coated with 100% and 75% AVG and analyzed for physical (color and texture), chemical (pH, TVB-N, TBARS, and PV), and sensory changes. Results showed that AVG significantly reduced lipid oxidation, as indicated by lower peroxide (PV) and thiobarbituric acid reactive substances (TBARS) values in treated samples compared to controls. While sensory, color, and texture parameters remained consistent across all groups, 75% AVG-coated slices extended the shelf life by four days, and the 100% AVG treatment achieved a two-day extension, based on TVB-N values. These findings highlight AVG’s potential as a natural, eco-friendly alternative to synthetic antioxidants for aquatic product preservation.

## 1. Introduction

Seafood is a great source of omega-3 polyunsaturated fatty acids (PUFAs), which are crucial for human nutrition. The prevalence of PUFAs in seafood generally makes it more susceptible to lipid oxidation. Formation of rancid odor and flavor, shortening of shelf life, loss of nutrition, color changes, and possible production of harmful compounds are the primary consequences of lipid oxidation in seafood [1]. European sea bass (*Dicentrachus labrax,* Linnaeus 1758) is one of the most economically important species in the world. Sea bass is sold in many different forms, such as whole fish, gutted or not, and sliced, ready-to-use flesh. The high demand for sea bass is due to its white color, sweet and mild flavor, and low-fat content. After capture, either wild or cultured, sea bass spoilage depends on the storage method [2,3]. To preserve its freshness, iced storage is commonly used, particularly during transportation for local distribution. With the increasing demand for ready-to-cook meals, sea bass fillets and slices are now widely available in supermarkets [4]. However, their perishability remains a challenge for the seafood industry. Lipid oxidation plays an important role in the quality of sea bass. During storage, lipid oxidation may result in the accumulation of harmful substances, which may lead to increased consumption risk [5]. Therefore, various preservation techniques have often been used in the past to delay lipid oxidation in seafood, such as synthetic antioxidants [6].

The use of butylated hydroxytoluene (BHT), butylated hydroxyanisole (BHA), and tertiary butylhydroquinone (TBHQ), known as synthetic antioxidants, is well known for effectively delaying lipid oxidation. Although synthetic antioxidants are successful in extending shelf life and preserving the sensory qualities of products subject to strict regulation, concerns persist regarding their safety due to potential excessive or improper application, as certain combinations may enhance their toxic effects [7]. Research on animals has shown that high doses of synthetic antioxidants can be harmful, leading to DNA damage, genetic errors, and the development of cancerous tumors. Since humans and animals share many similar genetic codes and biological processes, they often respond to environmental exposures in comparable ways [8]. In recent years, there has been increasing concern about the use of synthetic antioxidants in fish feed ingredients and their potential carryover into farmed fish fillets, raising important food safety considerations [9]. Consequently, there is growing interest in natural preservatives as new strategies to prevent the oxidation of seafood due to the high demand for food products free of synthetic additives [6]. Moreover, their chemical diversity includes polyphenols and flavonoids, and natural preservatives like plant extracts, whether in the form of pure substances or standardized extracts, offer limitless potential for inhibiting microbial development [10].

*Aloe vera*, also known as *Aloe barbadensis Miller*, is a highly significant plant-based treatment that has been thoroughly studied for use in pharmaceutical, food, and cosmetic applications [11]. Several bioactive compounds (aloin, anthraquinones, flavonoids, saponin, and aloe-mannan) identified in *A. vera*, including vitamins, aminoacids, and polyphenols, may act as reducing agents, hydrogen donors, and singlet oxygen quenchers [12,13]. *Aloe vera* gel (AVG) exhibits potent antioxidant properties due to its rich composition of bioactive compounds, which effectively scavenge free radicals, reduce oxidative stress, and contribute to its potential health benefits, such as anti-cancer, anti-diabetic, anti-viral, and disease-preventive effects [14]. In recent years, *Aloe vera* has been increasingly utilized in the processing of both animal- and plant-based foods, as well as for direct consumption. It is commonly incorporated into food formulations in various forms, including gel, powder, extracts, juice, pulp, and mucilage, either alone or combined with bioactive compounds [15].

Due to these unique properties, *Aloe vera* gel (AVG) has been extensively studied for its ability to combat microbial contamination and extend the shelf life of perishable products. As a natural preservative, AVG provides an effective and eco-friendly alternative to synthetic antimicrobial agents, demonstrating strong inhibitory effects against Gram-positive and Gram-negative bacteria, as well as various fungal pathogens [16]. AVG possessing both antioxidant and antibacterial qualities makes it an excellent choice for use as a natural preservative [17]. AVG applied as an edible coating has been found to be effective in extending shrimp shelf life and led to the retention of fresh flesh quality during cold storage [18]. According to Tri Winarni et al. [19], 20% *A.* vera and 1.5% crown of god fruit (*Phaleria macrocarpa*) treatments were found to be the best treatments to reduce changes in sensory and microbial quality of Indian mackerel during storage at 4 °C. In addition, the use of AVG has been applied mostly on fresh and fresh-cut fruits, such as cherries [20], peaches [21], and apples [22]. More research is needed for additional information on the application of AVG in preserving perishable foods, especially fish. To date, most studies have focused on the application of AVG to prevent deterioration in fruits and vegetables. However, limited information is available regarding the use of AVG on freshly caught fish to inhibit lipid oxidation during cold storage. Furthermore, the prevention of lipid oxidation in fresh fish fillets has rarely involved the use of natural AVG without any chemical additives. This study addresses these gaps by exploring the potential of natural AVG as a safe and effective antioxidant for preserving the quality of fresh-cut fish, offering a novel and environmentally friendly approach in post-capture fish preservation. Therefore, the objective of this study was to investigate the effect of applying AVG as a coating on sea bass slices to determine the changes in quality characteristics (physical, chemical, and sensory) during cold storage (4 ± 1 °C at 13 days).

## 2. Materials and Methods

### 2.1. Materials

The materials used include *Aloe vera*, wild sea bass, methanol (CH_3_OH; Honeywell Riedel-de Haën, Seelze, Germany), 2,2-Diphenyl-1-picrylhydrazyl (DPPH; Sigma-Aldrich, St. Louis, MO, USA), hexane (C_6_H_14_; Merck, Darmstadt, Germany), cumene hydroperoxide (CPO; Sigma-Aldrich, St. Louis, MO, USA), 1,1,3,3-Tetraethoxypropane (TEP; Sigma-Aldrich, St. Louis, MO, USA), chloroform (Carlo-Erba Reagent, Milan, Italy). Other solvents and reagents were obtained from Merck, Darmstadt, Germany. All reagents and solvents were of analytical grade.

### 2.2. Preparation of Aloe vera Gel

*Aloe vera* gel (AVG) was obtained by extracting the green leaves through filleting based on the method of Maan et al. [11] with modification. Pure leaves of *A. vera* were obtained from 3-year-old plants grown in home pots, washed with tap water, and then rinsed with distilled water [23]. After washing, the leaves were prepared by cutting their bases (1 cm), tops (1–2 cm), and sharp spines using a sterile surgical scalpel. Aloin (a bitter and yellow-brown colored compound) was removed from the leaves by vertical immersion in a container for one hour [24]. The immersion time was maintained to ensure complete removal of aloin from the gel. After immersion, the outer cortex layer of the leaves was removed using a sharp knife, and the colorless gel part was carefully scraped out with a clean spoon. The gel was then mixed homogeneously in a glass beaker to ensure consistency. The resulting mixture was a pure, colorless gel, which was filtered to separate any remaining fibers.

### 2.3. Fish Preparation and Coating with AVG

Wild sea bass (*Dicentrarchus labrax*) caught by local fishermen using a line in Kefken, Black Sea, Türkiye, within 24 h were purchased from a local retail fish vendor providing fish degutted and ready for consumption as food in Izmit (Kocaeli, Türkiye). Twenty whole fish with an average weight and length of 576.25 ± 54.58 kg and 30.74 ± 0.79 cm, respectively, were brought to the laboratory in ice within an hour. Fish samples were prepared by first removing the head part, washing them with tap water, and then cutting them into small, 1 cm thick cross-sections (slices). Each of the twenty fish was divided into 6–7 pieces of approximately equal size, starting from the part between the adipose fin and the pectoral fin. Then, the surface of both sides of these slices was coated with 1 mL of AVG. The sea bass slices were randomly divided into three groups: the control group (not coated with AVG), 100%, and 75% concentration (with water solution) AVG-coated. The coated samples were then dried at room temperature for approximately 15 min on each side and packaged in polyethylene plastic for storage at 4 ± 1 °C for 13 days. Triplicate samples were taken on days 0, 3, 6, 8, 10, and 13 of storage, and all experiments were performed in parallel at each sampling interval.

### 2.4. Determination of DPPH Radical Scavenging Activity

The antioxidant capacity of freshly prepared 75% AVG and 100% AVG was measured (*n* = 6) using the method of Brand-Williams et al. [25] just before being used for fish coating. In glass test tubes, 0.1 mL of both concentrations of AVG was mixed with 3.5 mL of 6 × 10^−5^ mol/L methanolic DPPH solution, which was prepared by ≥99.9% methanol. The mixtures were then incubated in the dark in a closed chamber by also covering the test tubes with aluminum foil to prevent any exposure to light (minimizing potential photodegradation of DPPH) for 30 min and measured at 515 nm with a spectrophotometer (Shimadzu UV-1280, Shimadzu Corporation, Kyoto, Japan). A completely undiluted solution of the DPPH concentration used in this study was read as a control. The percent inhibition of AVG was then calculated using the following equation:(1)$$Inhibition\left(\%\right)=\frac{Absorbance\;of\;control-Absorbance\;of\;concentrate}{Absorbance\;of\;control}\times 100$$

### 2.5. Chemical Analysis of AVG-Treated Sea Bass Slices

Moisture, ash, and fat content of sea bass were determined only for day 0 samples. Moisture content was determined by AOAC 950.46 using the forced air drying method at 105 °C in an oven (Nüve FN 500, NÜVE, Ankara, Türkiye). For ash content, the homogenized samples were burned in a muffle furnace (Nüve MF 120, NÜVE, Ankara, Türkiye) at 550 °C until the color became white. The method followed by AOAC 920.153. Crude fat content was measured by AOAC 991.36 by solvent extraction method in a soxhlet system (Velp SER 148, Velp Scientifica, Usmate Velate, Italy), where n-hexane was used as a solvent [26]. The pH was measured using a digital pH meter (Thermo Scientific Orion 3-Star Benchtop, Thermo Fisher Scientific, Beverly, MA, USA) on a homogeneous mixture prepared by blending 1 g of fish sample with 10 mL of distilled water (1:10, *w*/*v*) following the method described by Abelti [27].

Total volatile basic nitrogen (TVB-N) was determined using the direct MgO method described by Woyewoda et al. [28]. Ten grams of homogenized fish sample and 2 g of MgO were diluted to a final volume of 300 mL and placed on a heater. The heater was set to bring the mixture to a boil within 10 min, and once boiling started, distillate collection was continued for 25 min. TVB-N was expressed as mg N/100 g of sample. Thiobarbituric acid reactive substances (TBARS) were determined by the method of Yagiz et al. [29]. The results were expressed as micromoles of malonaldehyde (MDA)/kg fish meat. Peroxide determination (primary lipid oxidation products) was performed using Raghavan and Kristinsson’s method [30]. A standard curve was prepared using cumene hydroperoxide (CPO). Lipid hydroperoxides were calculated as mmol CPO/kg fish meat.

### 2.6. Texture Profile Analysis (TPA)

The TPA was measured using standard-size fillets (2 cm × 1 cm × 1 cm; length × width × thickness) in the epaxial and hypaxial muscles of fish chunks using a texture analyzer (TA Plus; AMETEK Lloyd Instruments Ltd., Bognor Regis, West Sussex, UK). The specimens were placed flat on the analyzer’s platen with the longest side aligned parallel to the compression axis. They were then compressed at 30% of their initial height at a speed of 3 mm/min under the influence of a 50 N load cell. A cylindrical probe with a diameter of 10 mm was used during the measurements [31]. Parameters obtained from the texturing device include hardness, springiness, chewiness, gumminess, and cohesiveness.

### 2.7. Color Measurement

Color measurement was performed using a machine vision system as described by Pilavtepe-Celik et al. [32]. The system comprises a lightbox with polarized and non-polarized illumination, a computer with a digital camera connected to the system, and color analysis software. The lightbox (Engineering and CyberSolutions, Gainesville, FL, USA) was installed in the laboratory and contained two daylight fluorescent tubes (Lumichrome F15W1XX, color temperature 6500 K, color retention index 98, Lumiram, Larchmont, NY, USA). A 51% light transmission diffuser, covered on the sample side by a polarization sheet (Rosco, Stamford, CT, USA), separated the interior of the lightbox from the samples. The system was connected to a Nikon D7000 digital camera (Nikon Corporation, Tokyo, Japan) with the following settings: aperture: f/9, shutter speed: 1/6, and preset: 138. To obtain an image, the fish slices were removed from the polyethylene plastic and set in the lightbox (Figure 1). The samples were taken using the “two image” method [33]. LenseEye-NET Version 2.0.1.2 (Engineering and CyberSolutions, Gainesville, FL, USA) software was used to determine L* (lightness), a* (redness/greenness), and b* (yellowness/blueness) values as defined in the CIELAb system from acquired images. The color values for the pixels representing the surface of the fish slices were determined using the software as the average L*, a*, and b* values.

### 2.8. Sensory Analysis

The sensory quality of the cooked sea bass slices was evaluated using a 9-point hedonic scale (1 = Dislike extremely to 9 = Like extremely). A sensory score of four was considered as the threshold for consumer acceptability [5,19,34]. Sensory analysis was performed at Kocaeli University, Izmit Vocational School, Food Technology Laboratory with the participation of nine volunteer panelists from the department members and graduate students (All assessors were provided with written details about the study before the experiments and voluntarily signed a consent form to participate.). Seabass slices (Control, 100%, and 75% AVG) were cooked in a conventional oven (Arzum Cookart Diji, Arzum, İstanbul, Türkiye) for 15 min at 200 °C.

Within all cooking processes, no salt, spices, food additives, frying oil, or oil were used, and all samples were cooked under identical conditions. After the cooking process, the slices have been divided into plates based on their treatment groups (three slices in each plate; control, 100%, and 75% AVG) and blinded with 3-digit codes. The panelists were asked to evaluate the samples in terms of color, odor, taste, texture, and overall quality. The fish samples evaluated were safe for consumption. Panelists were informed not to evaluate the “Taste” parameter after Day 6. This decision was based on total volatile basic nitrogen (TVB-N) analysis results, which indicated that the acceptable limit of consumption was reached after Day 6.

### 2.9. Statistical Analysis

Experiments were performed in triplicate for each group of samples, and in particular instances, parallel test experiments were used to improve data accuracy. In order to assess the data, Minitab 19 (Minitab Inc., State College, PA, USA, 2020) was used as statistical software. Data are presented as mean ± standard deviation. The results were analyzed using one-way ANOVA, preceded by a homogeneity of variance test to support the assumption of equal variance, and followed by Tukey’s multiple comparison test to identify significant differences across storage days for each treatment (*p* < 0.05). Pearson correlation was used to determine the linear correlation between odor, the sensory parameter, and chemical quality parameters (TBARS, PV, and TVB-N).

## 3. Results and Discussion

### 3.1. DPPH Radical Scavenging Activity

In this study, the DPPH radical scavenging activity of freshly prepared 100% and 75% AVG concentrations was determined to be 72.6% and 61.4%, respectively, just before being used for fish coating. These values are comparable to the results of Soltanizadeh and Mousavinejad [18] and Hu et al. [23], who found that a 100% concentration of *A. vera* had oxidation inhibitory values of 72.19% (three-year-old *A. vera*) and 67.51%, respectively. The decrease in absorbance of the DPPH radical is due to the scavenging capability of *A. vera* extracts at different concentration stages [23]. This highlights that as the concentration increases, the absorbance value decreases. The antioxidant content of an extract increases with its concentration; consequently, the more the extract inhibits DPPH, the less DPPH remains.

### 3.2. Chemical Analysis

The moisture, ash, and fat content were determined on day 0 of storage to evaluate the proximate composition of sea bass. The moisture content of the control, 100% AVG, and 75% AVG groups was measured as 69.90 ± 1.68%, 71.48 ± 1.49%, and 67.34 ± 1.36%, respectively. The fat content of the control, 100% AVG, and 75% AVG groups was measured as 2.44 ± 0.63%, 3.10 ± 0.78%, and 3.09 ± 0.51%, respectively. The ash content of the control, 100% AVG, and 75% AVG groups was measured as 1.82 ± 0.40%, 1.34 ± 0.18%, and 1.02 ± 0.12%, respectively. According to the results, the ash content of the 75% AVG group was found to be significantly lower compared to that of the control samples (*p* < 0.05). However, no significant differences in fat and moisture content were identified among any of the samples (*p* > 0.05). The proximate composition of sea bass in this study is similar to the results of other researchers [3,35,36]. According to the scale proposed by Stansby [37], sea bass can be classified as a low-fat fish, with a lipid content of lower than 5%.

The results of the pH, TVB-N, PV, and TBARS analyses conducted on sea bass slices stored under cold conditions on days 0, 3, 6, 8, 10, and 13 of storage are presented in Table 1. In general, the pH of freshly caught fish muscle is roughly 7, whereas the pH of postmortem fish muscle varies between 5.5 and 7.1 depending on the season, species, and other circumstances [2]. In this study, the initial pH values for the control, 100% AVG, and 75% AVG were 6.82 ± 0.04, 6.87 ± 0.09, and 6.87 ± 0.02, respectively. These values are consistent with the initial pH values of sea bass, which were reported as 6.75 ± 0.06 and 6.81 ± 0.00 by Periago et al. and Cakli et al. [38,39]. During storage, a significant increase in pH values was observed in the control, 100% AVG, and 75% AVG samples, beginning from the 8th day of storage (*p* < 0.05). This increase occurred at a faster rate in the control samples during cold storage. Specifically, the control sample showed a significant increase by day 8, while the AVG-coated samples exhibited this change by day 10. According to Soltanizadeh and Mousavinejad [18], the increase in pH values in shrimp coated with AVG during storage is attributed to the enzymatic production of compounds, such as trimethylamine and dimethylamine, as well as the accumulation of volatile basic nitrogen.

Total volatile basic nitrogen (TVB-N), primarily consisting of ammonia and other amines, arises from the decomposition of proteins and non-protein nitrogenous substances, predominantly due to microbial activity. TVB-N is widely used as an indicator for fish deterioration [34]. Generally, TVB-N levels between 30–35 mg N/100 g are considered the acceptability limit for cold-stored fish [40]. The initial TVB-N values for the control, 100% AVG, and 75% AVG samples were 24.02 mg N/100 g, 23.60 mg N/100 g, and 23.80 mg N/100 g of fish, respectively (Table 1). Fuentes et al. [36], in their comparison of wild and cultured sea bass *(Dicentrarchus labrax*), reported TVB-N concentrations of 23.19 ± 1.01 mg N/100 g in wild fish and 24.99 ± 1.31 mg N/100 g in cultured fish. By the 6th day of storage, the control samples reached the consumable limit with a TVB-N value of 35.00 ± 1.39 mg N/100 g, which was significantly higher than the sea bass slices coated with 100% and 75% AVG (*p* < 0.05). In this study, while the control samples exceeded the TVB-N limit on the 6th day, the 100% and 75% AVG-coated sea bass slices remained below the limit until the 8th and 10th days, respectively. The delay in spoilage observed in AVG-coated samples is attributed to the bioactive and antibacterial components of AVG, which reduce microbial growth and slow deterioration [41]. Consistent with our findings, Tri Winarni et al. [19] observed a slower increase in TVB-N values in mackerel coated with 20% AVG during storage, highlighting the preservative effects of *Aloe vera* and the crown of the god fruit. Soltanizadeh and Mousavinejad [18] also demonstrated that shrimp stored at 4 °C for 7 days and coated with 75% and 100% AVG had lower TVB-N values compared to untreated samples, due to reduced bacterial and enzymatic activity. For many fish species, TVB-N values are known to increase progressively as spoilage advances [42,43]. Faisal et al. [44] investigated the preservative effect of an edible coating mixture containing liquid smoke from OPEFB and turmeric on mackerel and reported that TVB-N values followed an increasing trend similar to total plate count (TPC) results as spoilage progressed, while remaining lower than those of control samples. Similarly, Fan et al. [34] studied the effect of a tea polyphenol dip on extending the shelf life of silver carp and found that treated samples had lower TVB-N and TPC values compared to the control. The reduction in TVB-N values was attributed to the lower microbial counts. In line with these findings, certain constituents in AVG, such as saponins, acemannan, and anthraquinones, exhibit inhibitory effects on microbial growth comparable to those of common antibacterial chemicals and antibiotics [14,15]. In our study, the lower TVB-N values in AVG-treated samples can be attributed to these antimicrobial properties. These findings confirm that coating sea bass with AVG significantly reduces TVB-N values during storage at 4 ± 1 °C (*p* < 0.05), extending its shelf life.

Throughout the 13-day storage period, the formation of primary oxidation products in sea bass slices was monitored through peroxide value (PV) measurements (Table 1). A significant increase in PV was observed on the sixth day for all samples during cold storage, followed by a decline until the thirteenth day in samples coated with 100% and 75% AVG (*p* < 0.05). However, no significant change was detected in the control sample from the sixth to the thirteenth day. Lipids are prone to oxidation to produce unstable hydroperoxides and gradually produce secondary products. The decrease in PV values after day 6 can be explained by the decomposition of hydroperoxide to secondary lipid oxidation products, such as aldehydes, ketones, alcohols, hydrocarbons, etc. [32,45]. Protein oxidation could lead to the release of iron ions, which in turn accelerates fat oxidation [45]. On the 13th day of storage, the TVB-N value showed a significant increase in samples coated with 75% AVG. This increase, which is an indicator of protein oxidation, corresponds with a rise in the PV value on day 13, suggesting that the observed change was driven by the protein oxidation process. To ensure product quality and prevent oxidation, PV values should remain below 10–20 meq O_2_/kg oil [46]. The findings of this study confirmed that all samples remained within these recommended limits. Notably, the PV values of samples coated with 100% and 75% AVG were significantly lower than those of the control group (*p* < 0.05). This reduction in lipid oxidation can likely be attributed to the antioxidant properties of *Aloe vera*, which contains phenolic compounds known for their free radical scavenging ability. Previous studies have demonstrated that phenolic compounds, such as flavonoids and proanthocyanidins, exhibit strong antioxidant activity by neutralizing free radicals and interrupting oxidative reactions [12].

A significant increase in thiobarbituric acid reactive substances (TBARS) values was observed in all samples from day 0 to day 13 during cold storage (*p* < 0.05) (Table 1). The initial TBARS values (day 0) for the control and 100% and 75% AVG-coated sea bass slices were 0.348 ± 0.012, 0.348 ± 0.017, and 0.195 ± 0.047 mg MDA/kg fish, respectively. By the 13th day of storage, these values increased to 0.678 ± 0.021, 0.648 ± 0.082, and 0.343 ± 0.056 mg MDA/kg fish, respectively. TBARS values of all treatments were under the proposed limit of 1.5 mg MDA/kg fish [47]. TBARS values in the control and 100% AVG-coated samples continued to rise throughout storage. This trend aligns with findings from previous studies [32,48,49]. This increase in TBARS values during storage was attributed to the second stage auto-oxidation of unsaturated fatty acids, during which peroxides are oxidized to aldehydes and ketones [34]. It was observed that the secondary oxidation products in the sea bass slices covered with 75% AVG had a significantly lower value than the control and 100% AVG application (*p* < 0.05). In samples treated with 75% AVG, no significant increase was observed after the 3rd day of storage, and TBARS values remained stable until day 13. The significantly lower PV and TBARS values in the 75% AVG-coated samples compared to the control and 100% AVG-coated samples (*p* < 0.05) may be attributed to the dilution effect of added water during the preparation of 75% AVG. Water plays a protective and pro-oxidative role in lipid oxidation. Lipid oxidation follows a non-linear trend with water activity (a_w_). Oxidation rates are intermediate at very low a_w_ (~0), lowest at a_w_ = 0.2–0.3, and increase as a_w_ rises to 0.3–0.9. However, at a_w_ near 1.0, oxidation slows down again due to water’s protective effects [50]. As moisture content increases, water plays a protective role in reducing oxidation through two key mechanisms. It interacts with metal catalysts, modifying their coordination sphere and reducing their catalytic activity. Additionally, it forms hydrogen bonds with hydroperoxides, stabilizing them and preventing their decomposition through initiation reactions [51,52]. AVG, with a moisture content of 99.0–99.5% and a water activity (a_w_) of 0.98–0.99, is believed to contribute to lipid oxidation prevention when applied to sea bass slices [15,16,53,54]. The high a_w_ of AVG helps limit oxidative reactions, acting as a natural protective barrier. In the 75% AVG treatment, the additional water used for dilution resulted in a higher moisture content compared to the 100% AVG treatment, leading to a greater inhibitory effect on lipid oxidation.

### 3.3. Texture Measurements

A major problem in the fish industry is the fact that the meat of marine animals softens faster than other meats. It is important to determine the textural properties of fish during storage to assess quality changes and shelf life stability [18,55,56]. The hardness, springiness, chewiness, gumminess, and cohesiveness of cold-stored sea bass slices were evaluated using texture analysis (Table 2). The initial hardness level of 75% AVG-coated samples exhibited significantly lower values than both the control and 100% AVG-coated samples (*p* < 0.05). While hardness remained stable in the control and 100% AVG groups throughout storage (*p* > 0.05), the 75% AVG-coated samples showed a significant increase in hardness from day 0 to day 10, followed by a sudden decrease on day 13 (*p* < 0.05). This decrease at the end of the storage can be attributed to the high TVB-N values indicating microbial spoilage, which is similar to the results of Lan et al. [45]. However, during storage, no significant difference was observed in the hardness between the treatment groups. Springiness measures how well the fish muscle returns to its original shape after compression; cohesiveness refers to the internal bonding strength of fish muscle [45]. There were no significant differences in springiness and cohesiveness values between treatments or within each group throughout storage (*p* > 0.05). Chewiness is a key texture parameter that reflects the energy required to break down fish muscle during chewing. It is influenced by hardness, cohesiveness, and springiness. In line with our findings, chewiness showed no significant differences between treatments or within each group throughout storage (*p* > 0.05). In this study, AVG treatment did not significantly impact the textural properties of sea bass slices.

### 3.4. Color Analysis

Images of cold-stored sea bass slices were taken on days 0, 3, 6, 8, 10, and 13 of storage, and their color analysis results are presented in Table 3. The color changes were assessed based on the average L*, a*, and b* values. Throughout the storage period, no significant changes were observed in any of the treatments (control, 100% AVG, and 75% AVG) (*p* > 0.05). However, differences between treatments were noted on specific days: the L* value of the 75% AVG-coated slices was significantly higher than that of the control and 100% AVG on day 8, while the b* value in the control slices was lower than in the 75% and 100% AVG-coated samples on day 10 (*p* < 0.05). For sea bass slices, an increase in the L* value indicates a lightening of the meat color, whereas an increase in the b* value suggests a yellowing effect. No significant variations in a* values were observed between treatments or storage days. The absence of major color changes throughout storage suggests that AVG treatment did not negatively impact the color properties of sea bass slices.

### 3.5. Sensory Analysis

The sensory evaluation results of cold-stored sea bass slices coated with AVG are presented in Table 4. The findings indicate that there were no significant differences (*p* > 0.05) between the control, 100% AVG, and 75% AVG-coated samples throughout storage. The sensory parameters of the sea bass slices were within the acceptable range until day 13. However, a significant reduction in all sensory parameters was recorded for all treatments after day 6 (*p* < 0.05). This decline aligns with the natural progression of lipid oxidation and microbial spoilage, which are common in refrigerated fish products. Studies on silver carp and sea bass have reported a gradual decline in all sensory scores over time [5,34,45]. These findings are further supported by the chemical analysis results (TVB-N, TBARS, and PV) of our study, which showed a gradual increase during storage, indicating fish deterioration. This suggests that the application of AVG did not negatively affect the sensory attributes of sea bass and is therefore acceptable for consumer consumption.

### 3.6. Correlation Analysis

Table 5 reflects the correlation between chemical quality parameters and odor, the sensory parameter, for all treatment groups of cold-stored sea bass slices. The results showed a significantly strong negative correlation between odor and TVB-N values across all treatment groups (*p* < 0.05). For the lipid oxidation indicators, there was only a significant, strong negative correlation between the TBARS value and the odor parameter for the 100% AVG treatment. The findings revealed that during the storage procedure, oxidative protein degradation was the principal cause of ongoing deterioration of fish quality. Microorganisms and endogenous enzymes contribute to the protein degradation of seabass, generating reducing chemicals including amines that raise TVB-N [45]. Given that seabass is a low-fat fish, protein oxidation is likely to be the main reason for deterioration and the sharp decline in sensory scores, especially odor.

## 4. Conclusions

This study demonstrated that *Aloe vera* gel (AVG) serves as an effective natural antioxidant for preserving the quality of cold-stored sea bass (*Dicentrarchus labrax*) slices. The application of 100% and 75% AVG coatings significantly reduced lipid oxidation, as evidenced by lower peroxide values (PV) and thiobarbituric acid reactive substances (TBARS) in treated samples compared to the control. Moreover, the 75% AVG treatment extended the shelf life by four days, while the 100% AVG treatment resulted in a two-day extension, indicating that AVG effectively slows down spoilage processes. However, one limitation of this study is that the total phenolic content and specific phenolic compounds (e.g., aloin) in AVG were not measured, which weakens the persuasiveness of the proposed antioxidant mechanism. Future studies should quantify these compounds to better understand their contributions to AVG’s preservation effects. Throughout storage, color and texture parameters remained stable, with minor variations that did not negatively impact product quality. Sensory evaluation revealed no significant differences between AVG-coated and control samples, suggesting that AVG application does not alter consumer acceptability. Additionally, AVG-treated samples exhibited lower total volatile basic nitrogen (TVB-N) values, further confirming its role in delaying microbial and enzymatic degradation. Previous studies have shown that AVG coatings delay microbial activity, which correlates with reduced protein degradation and lower TVB-N levels. Future research should incorporate microbial analysis to confirm these effects. Overall, these findings highlight the potential of AVG as an eco-friendly, natural alternative to synthetic antioxidants for seafood preservation. Further research should explore optimized AVG formulations or combinations with other natural preservatives to enhance its efficacy in prolonging the shelf life of perishable aquatic products.

## Figures and Tables

**Figure 1 foods-14-01185-f001:**
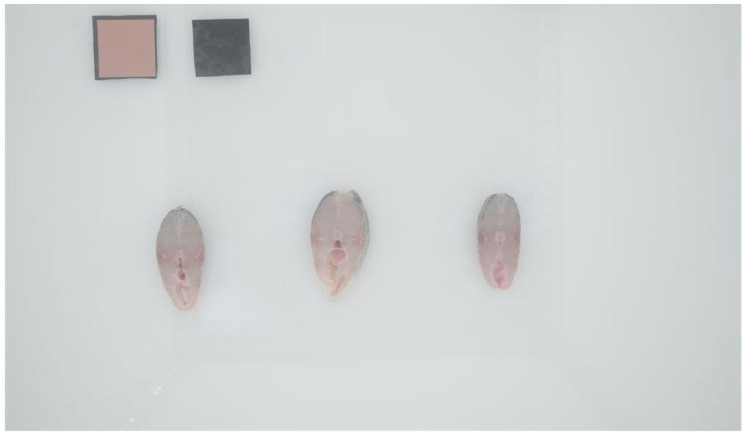
Image of fish slices acquired by machine vision system with the color reference and size reference (black square) within the lightbox.

**Table 1 foods-14-01185-t001:** Changes in pH, TVB-N, PV, and TBARS of cold-stored sea bass slices coated with AVG during storage.

Chemical Parameters	Days	Control	100% AVG	75% AVG
pH	0	6.82 ± 0.04 ^Aa^ *	6.87 ± 0.09 ^Aa^	6.87 ± 0.02 ^Aa^
	3	6.93 ± 0.09 ^Aa^	6.74 ± 0.39 ^Aa^	6.98 ± 0.09 ^Aa^
	6	6.99 ± 0.06 ^Aa^	7.02 ± 0.06 ^Aa^	7.06 ± 0.11 ^Aa^
	8	7.21 ± 0.24 ^Ab^	7.03 ± 0.21 ^Aa^	7.01 ± 0.02 ^Aa^
	10	7.32 ± 0.09 ^Ab^	7.45 ± 0.05 ^Ab^	7.28 ± 0.18 ^Ab^
	13	7.54 ± 0.06 ^Ac^	7.60 ± 0.18 ^Ab^	7.85 ± 0.12 ^Bc^
TVB-N (mg N/100 g fish)	0	24.02 ± 0.38 ^Aa^ *	23.60 ± 0.87 ^Aa^	23.80 ± 1.21 ^Aa^
	3	24.63 ± 0.41 ^Aa^	22.12 ± 0.81 ^Ba^	23.05 ± 0.65 ^Aba^
	6	35.00 ± 1.39 ^Ab^	31.35 ± 0.95 ^Bb^	25.18 ± 0.64 ^Ca^
	8	46.53 ± 0.68 ^Ac^	38.55 ± 3.36 ^Bc^	31.38 ± 1.01 ^Cb^
	10	59.36 ± 0.53 ^Ad^	48.50 ± 0.18 ^Bd^	44.22 ± 0.86 ^Cc^
	13	69.26 ± 0.23 ^Ae^	49.23 ± 0.20 ^Bd^	70.05 ± 0.18 ^Cd^
PV (mmol CPO/kg fish)	0	0.030 ± 0.018 ^Aa^ *	0.031 ± 0.020 ^Aa^	0.029 ± 0.016 ^Aab^
	3	0.084 ± 0.016 ^Ab^	0.075 ± 0.011 ^Ab^	0.044 ± 0.006 ^Bab^
	6	0.193 ± 0.009 ^Ac^	0.171 ± 0.030 ^Ac^	0.093 ± 0.027 ^Bc^
	8	0.193 ± 0.016 ^Ac^	0.078 ± 0.037 ^Bb^	0.042 ± 0.012 ^Bab^
	10	0.219 ± 0.011 ^Ac^	0.067 ± 0.001 ^Bab^	0.026 ± 0.009 ^Cb^
	13	0.199 ± 0.036 ^Ac^	0.061 ± 0.018 ^Bab^	0.055 ± 0.011 ^Ba^
TBARS (mg MDA/kg fish)	0	0.348 ± 0.012 ^Aa^ *	0.348 ± 0.017 ^Aa^	0.195 ± 0.047 ^Ba^
	3	0.491 ± 0.040 ^Ac^	0.447 ± 0.005 ^Aab^	0.371 ± 0.053 ^Bb^
	6	0.409 ± 0.021 ^Ab^	0.490 ± 0.050 ^Abc^	0.446 ± 0.126 ^Ab^
	8	0.449 ± 0.046 ^Abc^	0.525 ± 0.100 ^Abc^	0.319 ± 0.081 ^Bab^
	10	0.451 ± 0.012 ^Abc^	0.585 ± 0.074 ^Bcd^	0.315 ± 0.085 ^Cab^
	13	0.678 ± 0.021 ^Ad^	0.648 ± 0.082 ^Ad^	0.343 ± 0.056 ^Bb^

* Mean values ± standard error of means, *n* = 6. Means with different capital letters in a row are significantly different by treatments (*p* < 0.05). Means with different small letters in a column are significantly different by storage day (*p* < 0.05).

**Table 2 foods-14-01185-t002:** Changes in texture parameters of cold-stored sea bass slices coated with AVG.

Texture Parameters	Days	Control	100% AVG	75% AVG
Hardness (N)	0	13.82 ± 2.39 ^Aa^ *	11.82 ± 2.42 ^Aba^	5.62 ± 3.24 ^Ba^
	3	13.16 ± 2.00 ^Aa^	11.60 ± 2.20 ^Aa^	8.50 ± 2.00 ^Aab^
	6	15.25 ± 3.86 ^Aa^	12.51 ± 1.85 ^Aa^	10.11 ± 1.20 ^Aab^
	8	9.96 ± 2.60 ^Aa^	13.06 ± 1.24 ^Aa^	11.39 ± 1.64 ^Ab^
	10	11.31 ± 1.50 ^Aa^	12.66 ± 2.69 ^Aa^	12.28 ± 2.24 ^Ab^
	13	9.13 ± 1.53 ^Aa^	12.93 ± 4.79 ^Aa^	7.15 ± 0.26 ^Aab^
Springiness (cm)	0	0.60 ± 0.03 ^Aa^	0.66 ± 0.10 ^Aa^	0.76 ± 0.07 ^Aa^
	3	0.61 ± 0.02 ^Aa^	0.76 ± 0.22 ^Aa^	0.80 ± 0.07 ^Aa^
	6	0.68 ± 0.06 ^Aab^	0.71 ± 0.17 ^Aa^	0.82 ± 0.02 ^Aa^
	8	0.65 ± 0.05 ^Aab^	0.68 ± 0.10 ^Aa^	0.65 ± 0.04 ^Aa^
	10	0.72 ± 0.03 ^Aab^	0.74 ± 0.11 ^Aa^	0.70 ± 0.03 ^Aa^
	13	0.75 ± 0.08 ^Ab^	0.81 ± 0.20 ^Aa^	0.70 ± 0.10 ^Aa^
Chewiness (kgf.mm)	0	1.16 ± 0.11 ^Aa^	1.77 ± 1.68 ^Aa^	1.76 ± 0.38 ^Aa^
	3	1.38 ± 0.19 ^Aa^	1.67 ± 1.43 ^Aa^	1.22 ± 0.60 ^Aa^
	6	1.69 ± 0.37 ^Aa^	1.27 ± 0.37 ^Aa^	1.76 ± 0.86 ^Aa^
	8	1.00 ± 0.28 ^Aa^	1.51 ± 0.46 ^Aa^	1.18 ± 0.50 ^Aa^
	10	1.47 ± 0.25 ^Aa^	1.72 ± 0.84 ^Aa^	1.27 ± 0.36 ^Aa^
	13	1.15 ± 0.40 ^Aa^	1.90 ± 0.83 ^Aa^	1.13 ± 0.85 ^Aa^
Cohesiveness	0	0.14 ± 0.03 ^Aa^	0.54 ± 0.37 ^Aa^	0.20 ± 0.08 ^Aa^
	3	0.17 ± 0.01 ^Aa^	0.16 ± 0.06 ^Aa^	0.17 ± 0.07 ^Aa^
	6	0.16 ± 0.01 ^Aa^	0.14 ± 0.03 ^Aa^	0.20 ± 0.07 ^Aa^
	8	0.15 ± 0.03 ^Aa^	0.16 ± 0.02 ^Aa^	0.15 ± 0.04 ^Aa^
	10	0.18 ± 0.01 ^Aa^	0.17 ± 0.02 ^Aa^	0.14 ± 0.01 ^Aa^
	13	0.16 ± 0.02 ^Aa^	0.18 ± 0.02 ^Aa^	0.21 ± 0.13 ^Aa^

* Mean values ± standard error of means, *n* = 3. Means with different capital letters in a row are significantly different by treatments (*p* < 0.05). Means with different small letters in a column are significantly different by storage day (*p* < 0.05).

**Table 3 foods-14-01185-t003:** Changes in L*, a*, and b* values of cold-stored sea bass slices coated with AVG.

Color Parameters	Days	Control	100% AVG	75% AVG
L*	0	68.81 ± 1.17 ^Aab^ *	68.87 ± 0.58 ^Aa^	68.70 ± 1.05 ^Aa^
	3	70.34 ± 1.48 ^Aa^	66.46 ± 2.11 ^Aa^	68.58 ± 2.51 ^Aa^
	6	66.88 ± 0.68 ^ABab^	64.82 ± 0.99 ^Aa^	67.14 ± 0.97 ^Bb^
	8	66.75 ± 0.85 ^Ab^	66.58 ± 0.68 ^Aab^	69.88 ± 0.87 ^Bb^
	10	65.78 ± 1.62 ^Ab^	67.10 ± 1.26 ^Aab^	69.07 ± 2.65 ^Aa^
	13	67.90 ± 1.29 ^Aab^	68.07 ± 1.48 ^Aab^	68.04 ± 0.45 ^Aa^
a*	0	4.68 ± 0.44 ^Aa^	3.70 ± 0.72 ^Aa^	5.05 ± 1.20 ^Aa^
	3	3.49 ± 0.83 ^Aa^	3.39 ± 1.11 ^Aa^	4.51 ± 1.12 ^Aa^
	6	4.01 ± 0.42 ^Aab^	4.94 ± 0.78 ^Aa^	4.74 ± 0.78 ^Aa^
	8	4.63 ± 0.47 ^Aab^	4.59 ± 0.75 ^Aa^	3.93 ± 0.99 ^Aa^
	10	4.45 ± 0.15 ^Aab^	3.85 ± 0.42 ^Aa^	3.83 ± 0.33 ^Aa^
	13	5.14 ± 0.10 ^Ab^	4.55 ± 0.95 ^Aa^	4.53 ± 1.21 ^Aa^
b*	0	2.51 ± 0.83 ^Aa^	2.99 ± 0.91 ^Aa^	4.39 ± 0.97 ^Aa^
	3	4.34 ± 0.95 ^Aa^	3.74 ± 1.04 ^Aa^	3.32 ± 0.73 ^Aa^
	6	2.82 ± 1.65 ^Aa^	4.22 ± 0.41 ^Aa^	4.45 ± 1.06 ^Aa^
	8	4.10 ± 1.16 ^Aa^	3.70 ± 1.34 ^Aa^	4.67 ± 0.23 ^Aa^
	10	3.48 ± 0.80 ^Aa^	5.37 ± 0.10 ^Bb^	5.17 ± 0.74 ^Bb^
	13	5.04 ± 0.85 ^Aa^	5.02 ± 1.72 ^Aa^	4.95 ± 0.52 ^Aa^

* Mean values ± standard error of means, *n* = 3. Means with different capital letters in a row are significantly different by treatments (*p* < 0.05). Means with different small letters in a column are significantly different by storage day (*p* < 0.05).

**Table 4 foods-14-01185-t004:** Changes in sensory parameters of cold-stored sea bass slices coated with AVG.

Sensory Parameters	Days	Control	100% AVG	75% AVG
Color	0	7.67 ± 1.22 ^Aa^ *	7.78 ± 1.20 ^Aa^	8.22 ± 0.97 ^Aa^
	3	7.89 ± 0.60 ^Aa^	7.67 ± 0.50 ^Aa^	7.78 ± 0.44 ^Aab^
	6	7.78 ± 0.67 ^Aa^	7.56 ± 0.88 ^Aa^	7.78 ± 0.83 ^Aab^
	8	5.89 ± 1.36 ^Ab^	5.44 ± 1.51 ^Abc^	6.33 ± 1.58 ^Abc^
	10	6.67 ± 0.87 ^Aab^	6.56 ± 1.01 ^Aab^	6.56 ± 0.88 ^Ab^
	13	4.33 ± 0.71 ^Ac^	4.56 ± 1.42 ^Ac^	4.89 ± 1.27 ^Ac^
Odor	0	7.56 ± 1.42 ^Aa^	7.78 ± 1.39 ^Aa^	7.67 ± 1.66 ^Aab^
	3	8.00 ± 0.71 ^Aa^	7.00 ± 1.94 ^Aab^	7.89 ± 0.78 ^Aa^
	6	7.78 ± 0.67 ^Aba^	7.22 ± 0.67 ^Bab^	8.00 ± 0.50 ^Aa^
	8	4.44 ± 2.19 ^Ab^	3.89 ± 2.47 ^Ac^	4.00 ± 1.80 ^Ac^
	10	5.33 ± 0.71 ^Ab^	5.33 ± 0.71 ^Abc^	6.00 ± 1.00 ^Ab^
	13	2.33 ± 1.00 ^Ac^	3.00 ± 2.00 ^Ac^	3.00 ± 1.12 ^Ac^
Taste	0	7.89 ± 1.05 ^Aa^	7.33 ± 1.32 ^Aa^	7.67 ± 1.22 ^Aa^
	3	7.67 ± 0.71 ^Aa^	733 ± 1.41 ^Aa^	7.22 ± 2.11 ^Aa^
	6	7.67 ± 0.71 ^Aa^	6.67 ± 1.22 ^Aa^	7.78 ± 0.97 ^Aa^
	8	-	-	-
	10	-	-	-
	13	-	-	-
Texture	0	7.22 ± 1.72 ^Aa^	7.22 ± 1.09 ^Aa^	7.67 ± 1.41 ^Aa^
	3	7.56 ± 0.88 ^Aa^	7.78 ± 0.67 ^Aa^	7.78 ± 0.83 ^Aa^
	6	7.22 ± 1.20 ^Aa^	7.56 ± 1.13 ^Aa^	7.89 ± 0.93 ^Aa^
	8	5.89 ± 1.05 ^Aa^	6.11 ± 1.27 ^Aab^	6.44 ± 1.01 ^Aa^
	10	6.33 ± 1.00 ^Aa^	6.11 ± 1.62 ^Aab^	6.33 ± 1.32 ^Aa^
	13	3.78 ± 1.64 ^Ab^	4.56 ± 1.59 ^Ab^	4.44 ± 1.67 ^Ab^
Overall Quality	0	7.78 ± 1.20 ^Aa^	7.78 ± 0.67 ^Aa^	8.11 ± 0.78 ^Aa^
	3	7.67 ± 0.71 ^Aa^	7.22 ± 1.72 ^Aa^	7.33 ± 1.41 ^Aa^
	6	7.78 ± 0.83 ^Aa^	6.89 ± 1.05 ^Bab^	8.00 ± 0.87 ^Aa^
	8	4.78 ± 1.99 ^Ab^	4.33 ± 2.24 ^Ac^	4.78 ± 2.05 ^Abc^
	10	5.22 ± 1.30 ^Ab^	4.89 ± 1.36 ^Abc^	5.22 ± 1.39 ^Ab^
	13	2.56 ± 1.01 ^Ac^	3.11 ± 1.69 ^Ac^	3.00 ± 1.22 ^Ac^

* Mean values ± standard error of means, *n* = 9. Means with different capital letters in a row are significantly different by treatments (*p* < 0.05). Means with different small letters in a column are significantly different by storage day (*p* < 0.05).

**Table 5 foods-14-01185-t005:** Correlation of changes in odor and chemical parameters.

	Pearson Correlation Coefficients, *r* Value		
	ODOR		
Chemical Parameters	Control	100% AVG	75% AVG
TBARS	−0.762	−0.853 *	0.075
PV	−0.610	0.194	0.158
TVB-N	−0.913 *	−0.828 *	−0.806 *

* Significant correlation (*p* < 0.05).

## Data Availability

The original contributions presented in this study are included in the article. Further inquiries can be directed at the corresponding author.

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
