# Peer review of "Effect of Aloe vera Gel as a Natural Antioxidant on the Quality of Cold-Stored Sea Bass (Dicentrarchus labrax)"

_foods, 2025, doi:10.3390/foods14071185_

Round 1

Reviewer 1 Report

Comments and Suggestions for Authors

This manuscript (The Effect of Aloe vera Gel as a Natural Antioxidant on the Quality of Cold-Stored Sea Bass (Dicentrarchus labrax) is generally well-written. "The" at the beginning of the title can be removed. The stated objective is to investigate the effect of applying AVG as a coating on sea bass slices to determine the changes in quality characteristics (physical, chemical, and sensory) during cold storage (4±1°C at 13 days).   

Aloe vera gel has been used in the past to study the extension of shelf life of different foods, including seafood. So the authors should have demonstrated more novelty of this investigation. For, example, a case is made about the negative aspects of competing synthetic edible coatings or preservatives. One such product could have been tested even in a preliminary study (or a reference made) to see how they compare to make a stronger case for AVG.

The TPA section (2.6) should mention the sizes of fish used and manner of their placement on the platen described.

Lines 234-235. Please cite a particular reference or references to support the assertion.

Lines 262-269. It is intriguing that an increase in percent water can have such significant effect? The authors should have provided water activity data to make that case. The reference cited is about water content. So, what is the causative effect water activity or content?

Lines 283-285. We know that the way meat is cut can affect texture. So if it is true that the way that the fish was cut affected texture, why was data on alternative slicing mechanism presented to prove this suggestion?

Data and discussion of sensory parameters are presented. It would be interesting to know if  any correlations existed between odor and the other chemical quality parameters, especially PV.

Comparisons with literature data on shrimp was made. Are the spoilage products largely the same between these seafood types?

Lines 248-252. Did you determine the antioxidant activity (using the DPPH method) of AVG used in this study?

Author Response

The authors would like to sincerely thank the reviewer for their time and efforts to make this a better manuscript. Please find detailed responses as attached file and the corresponding revisions highlighted as yellow in the re-submitted manuscript.

Reviewer 2 Report

Comments and Suggestions for Authors The topic of this study holds practical significance, with a reasonable experimental design and relatively reliable data. However, the manuscript contains several deficiencies, including insufficient methodological details, inadequate mechanistic interpretation, linguistic irregularities, and inconsistent figure and table formatting. It is recommended that the authors address the following issues point-by-point and resubmit the revised manuscript to enhance its scientific value and publication potential.   1) The abstract lacks a clear statement of the research objective. The authors are advised to supplement the background and research purpose at the beginning of the abstract. 2) The rationale for selecting turbot as the experimental model has not been fully explained. The authors should elaborate on its commercial value and susceptibility to oxidation to justify the model selection. 3) The preparation method of aloe vera gel lacks specific parameters, such as the tool used for spine removal and the soaking time, which should be provided. 4) The DPPH assay does not specify the methanol concentration, light-shielding conditions, or the number of replicates (n value), which need to be clarified. 5) In the TVB-N determination, the amount of MgO used and the heating time were not reported, which should be supplemented. 6) The authors did not mention the homogeneity of variance test or the multiple comparison methods (such as Tukey or LSD) prior to ANOVA analysis, which should be clarified. 7) There is an inconsistency in the units of TVB-N in Table 1 and the main text. In Table 1, the unit is given as "mg N/100g," while some parts of the text use "mg/100g." The authors should standardize the unit expression. 8) In Figure 1, the light source parameters (such as color temperature and light intensity) were not indicated, which may affect the accuracy of color analysis. The authors are advised to provide a detailed description of these parameters. 9) The decimal places in the tables are inconsistent, for example, the pH values are expressed as "6,82±0,04" and "6.87±0.09." The authors should unify the decimal places according to a consistent format. 10) Although the TBARS value in the 75% AVG group was significantly lower than that in the control group, the PV value showed a rebound at a later stage (e.g., PV increased again on day 13). The authors should provide a reasonable explanation for this phenomenon. 11) The TVB-N value of the control group exceeded the acceptable limit on day 6, while the 75% AVG group remained within the acceptable range until day 10. The authors are advised to combine microbial data to explain this difference. 12) The study did not measure the total phenolic content or specific phenolic compounds (such as aloin) in the aloe vera gel, which weakens the persuasiveness of the proposed antioxidant mechanism. The authors are recommended to address this limitation. 13) The inhibitory effect of 75% AVG was superior to that of 100% AVG; however, the authors did not discuss whether excessively high aloe vera gel concentrations might induce hyperosmotic stress in fish muscles, which requires further interpretation. 14) The authors did not provide a clear explanation for the sharp decline in sensory scores after day 6, which should be discussed in greater depth. 15) The introduction and conclusion sections did not highlight the unique advantage of aloe vera gel, which possesses both antioxidant and antibacterial properties. The authors are suggested to emphasize this advantage to strengthen the scientific impact of the study.   Comments on the Quality of English Language

The English could be improved to more clearly express the research.

Author Response

The authors would like to sincerely thank the reviewer for their time and efforts to make this a better manuscript. We have carefully addressed all the comments and incorporated the necessary revisions into the manuscript. We provide detailed responses to each point, outlining the changes made as attached file. All modifications in the revised manuscript are highlighted as yellow for clarity.

Reviewer 3 Report

Comments and Suggestions for Authors

The manuscript described the effects of Aloe vera Gel on the quality of sea bass slices. Generally, the introduction is very weak and too short; it should claim the novelty of the current study. For example, what are the differences between other relative studies, such as "Aloe vera: A promising natural herbal supplement for enhancing growth, physiology, antioxidant activity and immunity in Catla catla" "Aloe gel polysaccharides and ascorbic acid promote collagen synthesis in mirror carp (Cyprinus carpio var. specularis)" and so on. Moreover, all the equations should be numbered (please refer to the template). Materials and methods should provide the references that are referred to. For example, the texture profile analysis should provide the reference. Please check all thereafter.

Lines 186-189: Percentage marks should be put after the numbers. Please confirm throughout the manuscript. 

Line 200: Please put a space between the number and the marks; please check throughout the manuscript. 

Table 1: It should be 6.82 ± 0.04 rather than "6,82 ± 0,04"; please revise throughout the manuscript. 

Lines 174-182: The discussion of DPPH is rather weak. The values of "72.6%" and "61.4%" belonged to day 13; where are the other data on 0, 3, 6, 8, and 10? In the material and method, it is not described clearly. 

Lines 275-286: the textural analysis should have data like chewiness, cohension and so on, where are other data? please discuss comprehensively. 

Author Response

The authors would like to sincerely thank the reviewer for their time and efforts to make this a better manuscript. Please find the detailed responses as attached file and the corresponding revisions highlighted as yellow in the re-submitted manuscript.

Round 2

Reviewer 2 Report

Comments and Suggestions for Authors

I have no problem.